computational chemistry/physical chemistry

electron correlation, correlation energy formulae, Hartree–Fock wavefunctions, Hartree–Fock critical nuclear charge, anion correlation energy

**Author for correspondence:**
Hazel Cox
e-mail: h.cox@sussex.ac.uk

This article has been edited by the Royal Society of Chemistry, including the commissioning, peer review process and editorial aspects up to the point of acceptance.

# Reparametrization of the Colle–Salvetti formula

Adam L. Baskerville, Msugh Targema and Hazel Cox

Department of Chemistry, School of Life Sciences, University of Sussex, Falmer, Brighton BN1 9QJ, UK

ALB, 0000-0002-0511-5689; MT, 0000-0003-4618-0521; HC, 0000-0001-5321-5417

We investigate the Colle–Salvetti (CS) formula, the basis of the Lee, Yang and Parr (LYP) correlation functional used in approximate density functional theory. The CS formula is reparametrized using high-accuracy Hartree–Fock (HF) wavefunctions to determine the accuracy of the formula to calculate anions. Fitting to the hydride ion or the two-electron system just prior to electron detachment at the HF level of theory does not, in general, improve the calculated correlation energies using the parameters derived from the CS/LYP method. An analysis of the CS parameters used in the popular LYP functional demonstrates the ingenuity and perhaps fortuitousness of the original formulation by CS.

## 1. Introduction

The Lee, Yang and Parr (LYP) correlation functional [1] is one of the most popular correlation functionals used in approximate density functional theory (DFT). It is used in over 40 different exchange–correlation functionals, including the ubiquitous B3LYP functional [2], to provide new knowledge and insight in a range of quantum chemistry applications.

The LYP correlation functional is based on the empirical correlation energy formula of Colle and Salvetti (CS) [3]. CS assumed the 'exact' wavefunction to be the product of a Hartree–Fock (HF) wavefunction and a correlation factor, which takes into account the correlation hole and was chosen to have the correct electron–electron cusp behaviour. Using this wavefunction, CS derived a correlation energy expression. Using the 1965 Clementi HF wavefunction for helium [4], CS fitted the correlation energy expression to a four-parameter function, $H(\beta, W)$, to reproduce the exact correlation energy of the He atom. LYP transformed this empirical correlation energy formula into an energy functional of the density, using the same four parameters derived by CS. Although there have been a number of critical analyses of the CS method (see in particular [5] or [6]) there is no doubt that the LYP correlation functional, combined with an appropriate exchange functional, has played a lead role in the success of Kohn–Sham DFT.

**Figure 1.** The $\{r, R, \theta\}$ coordinate system. The dashed lines represent the nucleus–electron coordinates $r_1$ and $r_2$.

Given the success of the LYP functional, the question arises whether using a significantly more accurate HF wavefunction in the fitting will influence the quality of the results produced. Recently, we published very accurate HF wavefunctions for two-electron atoms [7]. Therefore, to address this question we determine the CS fit parameters $\{a, b, c, d\}$ using our high-accuracy HF helium wavefunction. However, it is known that anions are particularly difficult to calculate. In fact it has been said that anions owe their stability to electron correlation effects [8] and this is certainly true for the hydride ion [9]. Furthermore, there have been significant efforts to assess DFT methods for the prediction of the electronic structures of complex and multiply charged anions (e.g. [10]). A particular criticism of the CS model is that the correlation hole is too short ranged and is thus biased towards regions of large electron density [6]. Recently, we have calculated and quantified the Coulomb holes of some heliogenic systems and shown that the Coulomb hole for a two-electron anion is significantly larger (approx. 3 times larger for the hydride ion compared with the helium atom) and the long-range behaviour of the intracule density [11] and radial density [12] of the anions is quite different to that of the helium atom.

Therefore, the purpose of this paper is twofold. The primary goal is to determine whether fitting the parameters in the CS formula using an HF wavefunction and correlation energy that captures the long-range low-density behaviour of anions, improves the calculation of electron correlation energies for anionic systems generally. Applications of anions are numerous in physics and chemistry from semiconductor technology [13] and solar cells [14] to mixed-anion compounds [15], making their accurate calculation essential; along with being able to accurately predict properties of new, exciting chemical phenomena. Initially, however, we determine whether the calculation of correlation energies can be improved by reparametrization of the CS formula using a very accurate helium HF wavefunction, which satisfies the exact conditions of the nucleus–electron cusp and Virial condition in addition to providing an energy that is orders of magnitude more accurate than that used in the original work.

# 2. Material and method

## 2.1. The Colle–Salvetti formula and Lee, Yang and Parr correlation functional

### 2.1.1. The Colle–Salvetti formula [3]

CS assumed that the correlated wavefunction of a closed-shell system can be approximated, as the product of a one-determinant HF wavefunction $\Psi_{\mathrm{HF}}$ and a Jastrow factor [16], attempting to correct for the missing electron correlation behaviour inherent in HF theory, i.e.

$$\Psi_{\mathrm{exact}}(\mathbf{x}_1, \mathbf{x}_2, \ldots, \mathbf{x}_N) = \Psi_{\mathrm{HF}}(\mathbf{x}_1, \mathbf{x}_2, \ldots, \mathbf{x}_N) \prod_{i>j}(1 - \phi(\mathbf{r}_i, \mathbf{r}_j)), \tag{2.1}$$

where $\mathbf{x}_i$ indicates all spatial and spin coordinates of electron $i$. The function $\phi(\mathbf{r}_i, \mathbf{r}_j)$ is chosen to be

$$\phi(\mathbf{r}_i, \mathbf{r}_j) = \exp(-\beta^2 r^2)\left(1 - \Phi(\mathbf{R})\left(1 + \frac{r}{2}\right)\right), \tag{2.2}$$

where $\mathbf{R} = \frac{1}{2}(\mathbf{r}_i + \mathbf{r}_j)$ is the extracular coordinate and $r = |\mathbf{r}_i - \mathbf{r}_j|$ the intracular coordinate [17,18], see figure 1. This choice of function enforces the electron–electron cusp condition and $\beta$ is related to the inverse of the radius of the Coulomb hole [19,20] which CS deduce to have the form

$$\beta = q\rho(\mathbf{R})^{1/3}, \tag{2.3}$$

by assuming that the correlation hole is proportional to the Wigner exclusion volume [21]; $q$ is an empirical parameter that determines the electron correlation length which CS calculated to be $q = 2.29$ for the helium atom and $\rho(\mathbf{R})$ is the electron density.

The correlation energy, $E_{corr}$, is given in terms of the diagonal HF spinless second-order density matrix, after making the assumptions that the second-order density matrix can be expressed in terms of the HF analogue and correlation corrections and that the first-order density matrix is equal to its HF analogue, i.e.

$$E_{corr} = \frac{1}{2} \int \int P_{2HF}(\mathbf{r}_1, \mathbf{r}_2)(\phi^2(\mathbf{r}_1, \mathbf{r}_2) - 2\phi(\mathbf{r}_1, \mathbf{r}_2))\frac{1}{r}\, d\mathbf{r}_1\, d\mathbf{r}_2. \tag{2.4}$$

After a number of approximations, and considering that at each point $\mathbf{R}$, $\Phi$ depends only on the electron density $\rho(\mathbf{R})$ through $\beta$, they found a simple approximation for $\Phi(\mathbf{R})$, i.e.

$$\Phi(\mathbf{R}) = \frac{1}{1 + \pi^{-1/2}/\beta}. \tag{2.5}$$

As $\beta$ is proportional to $\rho^{1/3}(\mathbf{R})$ (through the constant $q$) the extent of the correlation hole is related to $\rho^{-1/3}$; if $R$ is small the hole is small and of small extent and if $R$ is larger the hole is larger corresponding to the low-density situation [22].

CS then expand the second-order HF density matrix $P_{2HF}(\mathbf{r}_1, \mathbf{r}_2)$ to second order in $r$ around $r = 0$ [20], i.e.

$$P_{2HF}(\mathbf{r}_1, \mathbf{r}_2) = P_{2HF}(\mathbf{R} + \frac{\mathbf{r}}{2}, \mathbf{R} - \frac{\mathbf{r}}{2}) \approx P_{2HF}(\mathbf{R}, \mathbf{R}) + \frac{1}{6}[\nabla_r^2 P_{2HF}(\mathbf{r}, \mathbf{R})]r^2. \tag{2.6}$$

Substituting this in place of the two-electron density matrix, CS arrived at the following form to approximate the electron correlation energy, which they approximate using an analytic expression $H(\beta, W)$, i.e.

$$
\begin{aligned}
E_{corr} = &-\frac{1}{2} \int P_{2HF}(\mathbf{R}, \mathbf{R}) \int \frac{P_{2HF}(\mathbf{r}_1, \mathbf{r}_2)}{P_{2HF}(\mathbf{R}, \mathbf{R})} \left(2\exp(-\beta^2 r^2)\left(1 - \Phi(\mathbf{R})\left(1 + \frac{r}{2}\right)\right)\right. \\
&\left. - \exp(-2\beta^2 r^2)\left(1 - \Phi(\mathbf{R})\left(1 + \frac{r}{2}\right)\right)^2\right)\frac{d\mathbf{r}}{r}\, d\mathbf{R} \\
= &-\frac{1}{2}\int P_{2HF}(\mathbf{R}, \mathbf{R})\frac{4\pi}{\rho(\mathbf{R})}H(\beta, W)d\mathbf{R},
\end{aligned}
\tag{2.7}
$$

where $\rho(\mathbf{R})$ is the one-electron density, $P_{2HF}(\mathbf{r}_1, \mathbf{r}_2)$ is the two-electron density matrix and $P_{2HF}(\mathbf{R}, \mathbf{R}) = \rho(\mathbf{R})^2/2$. The function $H(\beta, W)$ is used to approximate the inner integral in $\mathbf{r}$ where

$$H(\beta, W) = a\left(\frac{1 + bW\exp(-c/\beta)}{1 + d/\beta}\right). \tag{2.8}$$

The parameters $\{a, b, c, d\}$ are determined by evaluating the inner integral for a set of $R$ values using the helium 1s HF orbital. The parameters derived in the fitting are: $a = 0.01565$, $b = 0.173$, $c = 0.58$, $d = 0.8$, which are used in the final CS formula for the calculation of the correlation energy using the electron density, $\rho(\mathbf{R})$, of the chosen closed-shell system, i.e.

$$E_{corr}^{CS} = -\pi \int \underbrace{a\left(\frac{1 + bW\exp(-c/\beta)}{1 + d/\beta}\right)}_{H(\beta, W)} \rho(\mathbf{R})d\mathbf{R}, \tag{2.9}$$

where $W$ is defined as

$$W = \frac{2}{q^2}\rho(\mathbf{R})^{-8/3}\nabla_r^2 P_{2HF}\left(\mathbf{R} - \frac{\mathbf{r}}{2}; \mathbf{R} + \frac{\mathbf{r}}{2}\right), \tag{2.10}$$

and $\nabla_r^2 P_{2HF}(\mathbf{R} - (\mathbf{r}/2); \mathbf{R} + (\mathbf{r}/2))$ is the Laplacian of the two-electron density matrix, which for two-electron atoms has the form [23]

$$\nabla_r^2 = \frac{1}{4}\rho(\mathbf{R})\left[\frac{d^2\rho(\mathbf{R})}{dR^2} + \frac{2}{R}\frac{d\rho(\mathbf{R})}{dR} - \frac{1}{\rho(\mathbf{R})}\left(\frac{d^2\rho(\mathbf{R})}{dR}\right)^2\right]. \tag{2.11}$$

Thus in deriving the formula (2.7), the correlation energy determines the contribution of the correlation hole surrounding every point $\mathbf{R}$, which depends upon the size of the hole through the Wigner radius $\rho^{-1/3}(\mathbf{R})$ [22]. CS used (2.9), with the parameters $a = 0.01565$, $b = 0.173$, $c = 0.58$, $d = 0.8$, to determine

the correlation energies of six atoms (He, Li$^+$, Be$^{2+}$, Be, B$^+$, Ne) and two molecules (CH$_4$, H$_2$O), reporting an average error of 2.5% and a highest error of 8%.

### 2.1.2. The Lee, Yang and Parr functional [1]

The CS formula is valid for closed-shell systems and LYP started by replacing the closed-shell diagonal density matrix by its equivalent open-shell form and converting the Laplacian of $P_{2HF}$ into one involving the Weizsacker kinetic energy density $t_W(\mathbf{R}) = \frac{1}{8}(|\nabla\rho(\mathbf{R})|^2/\rho(\mathbf{R})) - \frac{1}{8}\nabla^2\rho(\mathbf{R})$ and the HF kinetic energy density $t_{HF} = \frac{1}{8}\sum_i(|\nabla\rho(\mathbf{R})|^2/\rho_i(\mathbf{R})) - \frac{1}{8}\nabla^2\rho(\mathbf{R})$, to obtain the expression

$$E_{corr}^{CS-LYP}$$
$$= -a\pi \int \frac{\rho(\mathbf{R}) + (8b/q^2)\rho^{-5/3}(\mathbf{R})[\rho_\alpha(\mathbf{R})t_{HF}^\alpha + \rho_\beta(\mathbf{R})t_{HF}^\beta - \rho(\mathbf{R})t_W(\mathbf{R})]e^{-(c/q)\rho^{-1/3}(\mathbf{R})}}{1 + (d/q)\rho^{-1/3}(\mathbf{R})} \gamma(\mathbf{R})d\mathbf{R}, \tag{2.12}$$

where

$$\gamma(\mathbf{R}) = 2\left[1 - \frac{\rho_\alpha^2(\mathbf{R}) + \rho_\beta^2(\mathbf{R})}{\rho^2(\mathbf{R})}\right]. \tag{2.13}$$

In [1, eqn 10] the constants $\pi$ and $q$ appearing in (2.9) and (2.12) have been absorbed, such that $a^{LYP} = a\pi$, $b^{LYP} = 4b/q^2$, $c^{LYP} = c/q$, $d^{LYP} = d/q$. Equation (2.12) is equivalent to the closed-shell form, (2.9), when $\rho_\alpha(\mathbf{R}) = \rho_\beta(\mathbf{R}) = \rho(\mathbf{R})/2$, where $\rho_\alpha(\mathbf{R})$ and $\rho_\beta(\mathbf{R})$ are the $\alpha$-spin and $\beta$-spin electron densities [1].

To convert the CS energy formula (2.9) into an explicit functional of the electron density, LYP transformed the HF kinetic energy density into a pure density functional by performing a gradient expansion on $t_{HF}$ about the Thomas–Fermi local kinetic energy density, $t_{TF}$ [1]. The correlation energy formula of (2.12) then becomes

$$E_{corr}^{LYP} = -a\pi \int \frac{\gamma(\mathbf{R})}{1 + (d/q)\rho^{-1/3}(\mathbf{R})}\left\{\rho(\mathbf{R}) + \frac{8b}{q^2}\rho^{-5/3}(\mathbf{R})\left[2^{2/3}C_F\rho_\alpha^{8/3}(\mathbf{R})\right.\right.$$
$$+ 2^{2/3}C_F\rho_\beta^{8/3}(\mathbf{R}) - \rho(\mathbf{R})t_W + \frac{1}{9}(\rho_\alpha(\mathbf{R})t_W^\alpha(\mathbf{R}) + \rho_\beta(\mathbf{R})t_W^\beta(\mathbf{R})) \tag{2.14}$$
$$\left.\left.+ \frac{1}{18}(\rho_\alpha(\mathbf{R})\nabla^2\rho_\alpha(\mathbf{R}) + \rho_\beta(\mathbf{R})\nabla^2\rho_\beta(\mathbf{R}))\right]e^{-(c/q)\rho^{-1/3}(\mathbf{R})}\right\}d\mathbf{R}.$$

LYP used (2.14), to determine the correlation energies of the same set of molecules as CS, along with some open-shell atoms. They showed that their formulae give correlation energies within a few per cent of the experimentally determined values, in agreement with the original CS formula.

## 2.2. Implementation of the Colle–Salvetti method

In this work, the inner integral in the first expression for $E_{corr}$ in equation (2.7) is numerically integrated for a range of discrete $R$ values using an HF wavefunction (either the Clementi He wavefunction used by CS or using a very accurate two-electron wavefunction described in the next section), forming the data used to fit the function $H(\beta, W)$. The inner integral is calculated using the following form:

$$\int_0^\infty \int_0^\pi \frac{P_{2HF}(\mathbf{r}_1, \mathbf{r}_2)}{P_{2HF}(\mathbf{R}, \mathbf{R})}\left(2\exp(-\beta^2 r^2)\left(1 - \Phi(\mathbf{R})\left(1 + \frac{r}{2}\right)\right)\right.$$
$$\left.- \exp(-2\beta^2 r^2)\left(1 - \Phi(\mathbf{R})\left(1 + \frac{r}{2}\right)\right)^2\right)2\pi r^2 \sin(\theta)\frac{\rho(\mathbf{R})}{4\pi}d\theta\frac{d\mathbf{r}}{r}\bigg|_{R=R'} \tag{2.15}$$

where $R$ is fixed to a value $R'$ while integrating over $r$, $\theta$. The term $2\pi r^2\sin(\theta)$ originates from the Jacobian factor, and if integrating over all $\{r, R, \theta\}$, the volume element has the form

$$d\tau = 8\pi^2 r^2 R^2 \sin(\theta)drdRd\theta. \tag{2.16}$$

Here we conduct the multi-dimensional integration using the cuhre cubature numerical integration algorithm sourced from the C++ CUBA library [24]. The following coordinate transformations are

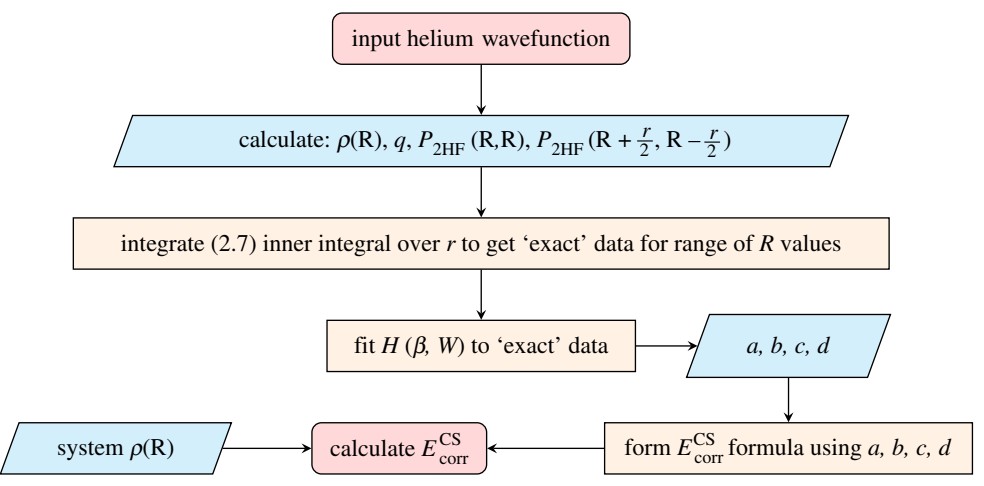

**Figure 2.** Flow diagram of the CS process used to determine the $E_{corr}^{CS}$ formula and hence calculate electron correlation energies. In this work, we use alternative two-electron wavefunctions as input and use the derived $\{a, b, c, d\}$ to calculate $E_{corr}^{CS-LYP}$ and $E_{corr}^{LYP}$.

applied to transform the inter-particle coordinates $\{r_1, r_2, r_{12}\}$ [7], into $\{r, R, \theta\}$, see figure 1,

$$r_1 = \sqrt{\frac{r^2}{4} + rR\cos(\theta) + R^2},$$

$$r_2 = \sqrt{\frac{r^2}{4} - rR\cos(\theta) + R^2} \qquad (2.17)$$

and

$$r_{12} = r.$$

Following this procedure allows the inner integral to be approximated using the function form $H(\beta, W)$, equation (2.8), resulting in the final CS expression for approximating the electron correlation energy of a given system.

The parameters $\{a, b, c, d\}$ are calculated by minimizing the square difference between the data calculated using the numerical integration of equation (2.15) and $H(\beta, W)$. The least-squares fitting is performed using either the Python SciPy.optimise.minimize function [25] or the high accuracy Levenberg–Marquardt algorithm [26] to minimize the square difference. These algorithms use different methodologies to optimize the parameters, where the latter provides, in general, a tighter fit. A convergence tolerance on each parameter of $1 \times 10^{-6}$ was used after it was found that a tighter fit could be problematic.

Integration over $R$ in equations (2.12) and (2.14) is implemented using a 100 point Gauss–Chebyshev grid [27] where a logarithmic transformation is applied [28], transforming the integration range $[-1, 1]$ to the required integration range $[0, \infty)$ valid for $R$.

Our exploration is focused on the CS methodology, but the LYP expression (2.14) is also used to determine correlation energies. To calculate $E_{corr}$ for systems other than the two-electron systems used in the derivation of the parameters, the HF wavefunctions from Koga *et al.* are used [29].

To clarify the multistep CS process, figure 2 depicts a flow diagram summarizing the key steps to arrive at the approximate CS expression for the electron correlation energy.

## 2.3. Deriving the high-accuracy Hartree–Fock wavefunctions

Accurate HF Laguerre-based wavefunctions were derived for the helium atom, the hydride ion and HF critical nuclear charge system $Z_C^{HF}$ using the method reported in [7]. In brief, the HF wavefunction, $\psi_{HF}$, is taken as the product

$$\psi_{HF}(r_1, r_2) = \psi(r_1)\psi(r_2), \qquad (2.18)$$

where the spin, which embeds the anti-symmetry of the total wavefunction, has been integrated out. The one-electron orbitals $\psi(r_i)$, have the form

$$\psi(r_i) = e^{-(1/2)Ar_i} \sum_{n=0}^{\infty} C(q)L_n(Ar_i), \quad i = 1 \text{ or } 2, \qquad (2.19)$$

**Table 1.** Energy (a.u.), expectation values (a.u.) and cusp values for He, $Z_C^{HF}$ and H$^-$ at the HF level of theory. The energy and inter-particle expectations values are accurate to the number of digits presented. The exact value of the nucleus–electron cusp, $v_{31}$ is $-Z$ and the exact value of the Virial condition, $\eta = |\langle \hat{V} \rangle / \langle \hat{T} \rangle + 2|$ is 0.

| property | He | $Z_C^{HF}$ | H$^-$ |
|---|---|---|---|
| energy | $-2.8616799956122$ | $-0.531663547021$ | $-0.48792973437$ |
| $\langle r_1 \rangle$ | 0.92727340473149 | 2.372691817 | 2.50395963 |
| $\langle r_{12} \rangle$ | 1.36212438367607 | 3.537395413 | 3.73927400 |
| $\langle \delta(r_1) \rangle$ | 1.7979591 | 0.1734895 | 0.15459 |
| $\langle \delta(r_{12}) \rangle$ | 0.1906039978065 | 0.0148160975759 | 0.012983476397 |
| $\langle 1/r_1 \rangle$ | 1.687282215 | 0.7170563797 | 0.685672155 |
| $\langle 1/r_{12} \rangle$ | 1.02576886989955 | 0.415497756084 | 0.39548484311 |
| $v_{31}$ | $-1.9999998$ | $-1.031180$ | $-1.000005$ |
| $\eta$ | $1.092 \times 10^{-22}$ | $7.169 \times 10^{-21}$ | $2.287 \times 10^{-20}$ |

and $r_1$ and $r_2$ are the nucleus–electron distances (figure 1). $L_n(x)$ is a Laguerre polynomial of degree $n$ and $A$ is a nonlinear variational parameter. The one-electron terms arising in the HF equations are solved using the series solution method (using Maple to generate the recursion relation) and the two electron integrals are solved analytically by exploiting the properties of the Laguerre polynomials [7] using an in-house Python code. The sum of the one-electron and two-electron matrix elements are used to create the Fock matrix, and the Fock equations are solved as a generalized eigenvalue problem, to determine new wavefunction coefficients. The convergence threshold for the SCF procedure was set at $1 \times 10^{-20}$ and was performed using direct inversion of iterative space (DIIS) and the variational parameter is optimized using the BOBYQA algorithm [30]. The accuracy of the implementation reported in [7] has been improved by using octuple, 64-digit precision and making use of ball arithmetic to enforce rigorous error bounds [31].

# 3. Results and discussion

Atomic units (a.u.) are used throughout, where $m_e = \hbar = (4\pi\epsilon_0)^{-1} = e = 1$ and the atomic unit of energy is the Hartree and the atomic unit of length is the Bohr.

## 3.1. The Hartree–Fock wavefunctions

A 25-term wavefunction was used, improving on the previously published data for He and H$^-$ [7]. The critical nuclear charge $Z_C$ is the minimum charge required for an atomic system to have at least one bound state. At the restricted HF level of theory, the hydride ion is unbound ([9] and references therein) and the critical charge for binding, corresponding to the energy at which the three-body system equals the lowest continuum threshold, is $Z_C^{HF} = 1.031177528$ [7]. Alternatively, the critical nuclear charge can be defined as the point at which the occupied orbital energy becomes zero, which corresponds to a value of $Z_C^{HF} = 0.828161008$ [32]. This system was also considered but did not provide an improved correlation energy formula, see electronic supplementary material, so is not discussed further. The wavefunctions derived for these four systems are provided in the electronic supplementary material.

The energy and the quality of the wavefunctions for the three systems: He, H$^-$ and $Z_C^{HF} = 1.031177528$, is reported in table 1. It can be seen that the energies are accurate to at least 11 significant figures and the wavefunctions are capable of determining accurate expectation values, provide accurate nucleus–electron cusp values, and satisfy the Virial condition extremely well.

## 3.2. Reparametrizing the Colle–Salvetti formula using the helium Hartree–Fock wavefunction

### 3.2.1. Fitting to helium atom data

The first step in deriving the parameters $\{a, b, c, d\}$ is to use the HF wavefunction to perform the integration (2.15) for a range of $R$ values. There are a number of considerations: (i) the range of $R$

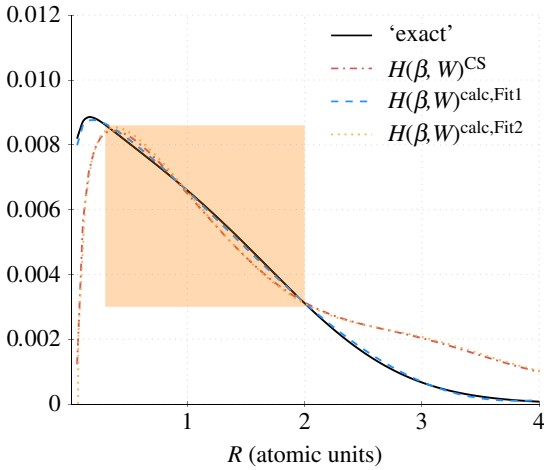

**Figure 3.** Plot of the 'exact' values (2.15) used to determine the parameters $\{a, b, c, d\}$ in the function $H(\beta, W)$ (see equation (2.7) and text for details). $H(\beta, W)^{CS}$ uses the parameters determined by CS. $H(\beta, W)^{calc,Fit\ 1}$ uses the 'Fit 1' parameters and $H(\beta, W)^{calc,Fit\ 2}$ uses the 'Fit 2' parameters determined in the present work. The shaded region highlights the range $0.3 \leq R \leq 2$.

values, (ii) the number of $R$ points to be used in the fitting (i.e. the step size), (iii) the optimizer and precision/tolerance for the least-squares fit to the function $H(\beta, W)$, and (iv) the initial guess values and possible boundary conditions on the parameters, given that this is a nonlinear least-squares fit; all these variables have been investigated [33].

The value of $q$ was taken to be 2.29, as in the original work of CS [3]. The integration was performed using both the Clementi HF wavefunction and the Laguerre-based wavefunction described in §3.1. The values using the new wavefunction agreed with the 10 points provided in [3, table 1] and offered an improved accuracy. These data points were then used to fit $H(\beta, W)$ to determine the new values of $\{a, b, c, d\}$. Figure 3 shows the 'exact' values obtained from the integration of (2.15) (arising from l.h.s. inner integral of (2.7)) and the $H(\beta, W)$ function (2.8) (see r.h.s. of (2.7)) using both the CS-derived values, i.e. $a = 0.01565$, $b = 0.173$, $c = 0.58$, $d = 0.8$, labelled $H(\beta, W)^{CS}$ and values obtained in this work $a = 0.01212$, $b = 0.03163$, $c = 0.11764$, $d = 0.74324$ obtained by fitting 80 $R$ values in the range 0.01–4 a.u. with guess values $a = b = c = d = 0.1$ using SciPy.optimise.minimize with a tolerance of $1 \times 10^{-6}$ without bounds on the parameters, labelled $H(\beta, W)^{calc,Fit1}$. The value of $\chi^2 = 0.000262$, demonstrating the very good-fit quality. This will be referred to as 'Fit 1' in the next section.

It is clear from figure 3 that the CS parameters that appear in the CS formula and LYP functional are a poor fit to the 'exact' data, whereas our derived parameters in $H(\beta, W)^{calc,Fit1}$ provide a very good fit to the 'exact' data capturing the maximum that occurs at 0.21 a.u.; this maximum has shifted to 0.36 a.u. in $H(\beta, W)^{CS}$.

The $R$ range and spacing were not specified by CS, but if we assume that they used just the 10 data points reported [3, table 1], which correspond to the range $0.3 \leq R \leq 2$, the $H(\beta, W)^{CS}$ function is a reasonable fit (see shaded region in figure 3). To further test the dependence of the fit quality on the range of $R$ values used in the fitting, a number of initial ($R_i$) and final ($R_f$) values were used to determine the parameter set $\{a, b, c, d\}$ using 100 $R$ values with the Levenberg–Marquardt algorithm [26] with the only bound on the parameters being they must be positive. A $\chi^2$ test was used to assess the fit quality.

The results in table 2 show that all the calculated fittings result in a better value of $E_{corr}^{calc}$ for the helium atom compared with CS but that the overall fit quality deteriorates when $R_f \geq 5$. It also shows that the $R$-range has little effect on the resulting correlation energy of helium, even when $c$ is numerically zero in the $0.1 \leq R \leq 2$ a.u. and $0.3 \leq R \leq 2$ a.u. cases; the $c$ parameter controls the exponential required to kill the divergence of the expression as $R$ increases.

A slightly improved CS fit over the range 0.3–2 a.u. was possible by starting the optimization from the CS parameters, imposing bounds on the parameters: $0 \leq \{a, b, c, d\} \leq 5$, and using the SciPy.optimise least-squares method. This dependence of the optimized fit parameters on the starting values is a common problem, see for example Chernov *et al.* [34]. This resulted in $E_{corr}^{calc} = -0.041925$, with the parameter values $a = 0.01628$, $b = 0.18438$, $c = 0.57594$, $d = 0.80562$. This will be referred to as 'Fit 2' in the next section and is shown in figure 3. The value of $\chi^2 = 0.001067$, demonstrating the reasonable but poorer fit compared with 'Fit 1'.

**Table 2.** The value of the correlation energy $E_{\text{corr}}^{\text{calc}}$ calculated from (2.12) using $\{a, b, c, d\}$ values obtained by fitting $H(\beta, W)$ to 100 'exact' data points obtained for $R_i \leq R \leq R_f$ and compared with the literature value. $\chi_{\text{calc}}^2$ represents the $\chi^2$ test of $H(\beta, W)^{\text{calc}}$ against the data points calculated using equation (2.15). $\chi^2$ calculated using the CS parameters in the range $0.3 \leq R \leq 2$ a.u. was 0.000860 and when calculated in the range $0.1 \leq R \leq 6$ a.u. was 0.912121.

| $R_i$ (a.u.) | $R_f$ (a.u.) | $a$ | $b$ | $c$ | $d$ | $E_{\text{corr}}^{\text{calc}}$ (a.u.) | $\chi_{\text{calc}}^2$ |
|---|---|---|---|---|---|---|---|
| 0.10 | 2 | 0.011 705 | 0.020 904 | $1.26 \times 10^{-11}$ | 0.718 962 | −0.042 203 568 | 0.000 027 |
| 0.10 | 3 | 0.011 942 | 0.029 122 | 0.091 153 | 0.724 517 | −0.042 293 055 | 0.000 140 |
| 0.10 | 4 | 0.012 045 | 0.033 531 | 0.122 470 | 0.721 393 | −0.042 330 509 | 0.000 368 |
| 0.10 | 5 | 0.012 119 | 0.026 835 | 0.099 306 | 0.774 389 | −0.042 230 030 | 0.040 234 |
| 0.10 | 6 | 0.012 183 | 0.025 119 | 0.094 061 | 0.797 276 | −0.042 188 053 | 0.383 999 |
| 0.30 | 2 | 0.011 485 | 0.024 418 | $3.11 \times 10^{-9}$ | 0.664 135 | −0.042 032 252 | 0.000 009 |
| 0.30 | 3 | 0.011 640 | 0.042 465 | 0.145 606 | 0.619 166 | −0.042 082 281 | 0.000 033 |
| 0.30 | 4 | 0.011 733 | 0.037 604 | 0.131 904 | 0.656 988 | −0.042 119 496 | 0.000 761 |
| 0.30 | 5 | 0.012 070 | 0.027 140 | 0.100 014 | 0.765 917 | −0.042 192 834 | 0.043 567 |
| 0.30 | 6 | 0.012 216 | 0.024 865 | 0.093 423 | 0.803 485 | −0.042 210 782 | 0.390 600 |
| CS: [3] | | 0.015 65 | 0.173 | 0.58 | 0.8 | −0.041 56 | |
| Exact: [7] | | | | | | −0.042 044 381 422 | |

### 3.2.2. Electron correlation energies

The key performance indicator of the function fits is how the parameters $\{a, b, c, d\}$ perform in calculating electron correlation energies for systems not used in the fit. Anions were not considered in the testing of the original CS function [3] or LYP functional [1], so we now test their accuracy when calculating electron correlation energies of atomic anions.

Table 3 reports electron correlation energies calculated using the CS formula (2.12) with (i) the CS values of $\{a, b, c, d\}$, (ii) our best-fit values labelled He (Fit 1) with $a = 0.01212$, $b = 0.03163$, $c = 0.11764$, $d = 0.74324$ obtained by fitting 80 $R$ values in the range 0.01–4 a.u. and without bounds imposed, (iii) best-relaxed fit values labelled He (Fit 2) with $a = 0.01628$, $b = 0.18438$, $c = 0.57594$, $d = 0.80562$ obtained by fitting 80 $R$ values in the range 0.3–2 a.u. with bounds on the parameters and starting close to the CS values.

Table 3 shows that a similar accuracy to CS can be obtained by following their approach, using a shorter $R$ range which does not capture the essential features of $H(\beta, W)$. Using an increased number of data points in the fitting and/or a more accurate HF wavefunction does not seem to influence the results significantly. Furthermore, it is very clear that a fit that accurately reproduces the 'exact' data over a more appropriate $R$ range performs badly for all other types of systems considered.

The results in table 3 suggest that the CS parameters give better correlation energies for the cationic and neutral atomic systems and that the relaxed fit parameters (Fit 2) are slightly better for the anions. It is clear that the best fit (Fit 1) (in terms of $\chi^2$-value and reproduction of the He correlation energy) performs extremely badly for all other systems considered.

Through accident or by design, the key feature of $H(\beta, W)$ derived by CS is the flexibility inherent in its shape. Figure 3 shows that the CS function fit, $H(\beta, W)^{\text{CS}}$, for helium is poor when considering the range $0 \leq R \leq 4$ a.u., but by relaxing the fit to the helium data and using a limited $R$ range, the function shape is applicable across a range of chemical systems.

To explore the fit to anions further and highlight any differences in the performance of the CS and LYP formulations, (2.12) and (2.14) are used to calculate electron correlation energies for a selection of atomic anions, table 4. Included are the exact electron correlation energies, $E_{\text{corr}}^{\text{exact}}$, calculated using Löwdin's definition [37], i.e. $E_{\text{corr}}^{\text{exact}} = E_{\text{exact}} - E_{\text{HF}}$. Here $E_{\text{HF}}$ represents the HF energies, taken from the high accuracy HF calculations of Koga *et al.* [29] with King *et al.* used for the hydride ion [7]. $E_{\text{exact}}$ is the exact ground state energy, calculated by adding the experimental electron affinity [36] to the estimated exact, non-relativistic energy of the neutral atom [35]. The percentage error between $E_{\text{corr}}^{\text{CS}}$ and $E_{\text{corr}}^{\text{exact}}$ is calculated using % Error $= ((E_{\text{corr}}^{\text{CS}} - E_{\text{corr}}^{\text{exact}})/E_{\text{corr}}^{\text{exact}}) \times 100$.

Table 4 reveals a range of accuracies when applying the CS and LYP functionals to atomic anions. Using the CS functional for systems with fewer than 11 electrons, the errors in calculated electron

**Table 3.** Correlation energies (a.u.) of atomic systems calculated using the CS fit parameters, our best He fit parameters (Fit 1) and He relaxed fit parameters (Fit 2) using (2.12), with the density matrices calculated using the HF wavefunctions of Koga *et al.* [29]. $E_{corr}^{Exact}$ are experimental estimates taken from [35] for cations/neutrals and [36] using the electron affinity for anions, except for two-electron systems which were calculated [7].

| type | system | $E_{corr}^{CS}$ | $E_{corr}^{Fit\ 1(Best)}$ | $E_{corr}^{Fit\ 2(Relax)}$ | $E_{corr}^{Exact}$ |
|---|---|---|---|---|---|
| *cations* | Li$^+$ | −0.043 884 | −0.048 845 | −0.043 838 | −0.043 498 |
| | Be$^+$ | −0.058 123 | −0.057 713 | −0.059 085 | −0.047 37 |
| | B$^+$ | −0.105 959 | −0.094 253 | −0.108 756 | −0.111 34 |
| | C$^+$ | −0.144 020 | −0.118 986 | −0.148 939 | −0.138 8 |
| | N$^+$ | −0.175 908 | −0.137 793 | −0.182 818 | −0.166 61 |
| | O$^+$ | −0.202 159 | −0.152 537 | −0.210 769 | −0.194 23 |
| | F$^+$ | −0.276 579 | −0.210 505 | −0.288 051 | −0.261 09 |
| average % error | | 6.8 | 17.4 | 9.1 | |
| *neutrals* | He | −0.041 560 | −0.042 353 | −0.041 925 | −0.042 044 |
| | Li | −0.050 302 | −0.051 928 | −0.050 877 | −0.045 33 |
| | Be | −0.092 596 | −0.084 308 | −0.094 840 | −0.094 34 |
| | B | −0.128 190 | −0.108 609 | −0.132 305 | −0.124 85 |
| | C | −0.160 596 | −0.128 410 | −0.166 654 | −0.156 4 |
| | N | −0.188 301 | −0.144 347 | −0.196 106 | −0.188 31 |
| | O | −0.261 061 | −0.200 030 | −0.271 775 | −0.257 94 |
| | F | −0.321 662 | −0.246 288 | −0.334 827 | −0.324 53 |
| | Ne | −0.375 313 | −0.285 952 | −0.390 819 | −0.390 47 |
| average % error | | 2.8 | 17.1 | 4.3 | |
| *anions* | H$^-$ | −0.030 724 | −0.027 513 | −0.031 398 | −0.039 821 282 |
| | Li$^-$ | −0.070 081 | −0.067 765 | −0.071 349 | −0.072 6 |
| | B$^-$ | −0.136 152 | −0.113 688 | −0.140 800 | −0.145 008 |
| | C$^-$ | −0.167 977 | −0.132 551 | −0.174 568 | −0.182 59 |
| | N$^-$ | −0.237 865 | −0.184 774 | −0.247 407 | −0.269 813 57 |
| | O$^-$ | −0.299 123 | −0.230 404 | −0.311 267 | −0.331 254 1 |
| | F$^-$ | −0.354 303 | −0.270 636 | −0.368 911 | −0.399 53 |
| average % error | | 10.3 | 25.8 | 7.3 | |

correlation energies can be substantial, e.g. greater than 22% for H$^-$. For systems with greater than or equal to 11 electrons, the error in calculated electron correlation energy stabilizes to approximately 1–2%. The LYP functional performs slightly better than the CS functional for anions with fewer than 11 electrons, but slightly worse for anions with greater than or equal to 11 electrons. The errors also stabilize for systems with greater than or equal to 11 electrons for the LYP functional.

It is not unexpected that the CS and LYP functionals calculate a greater than 20% error in the electron correlation energy for the hydride ion, given that long-range correlations such as those present in the hydride ion play little role in recovering the correlation energy within the CS method [5,6]. However, what is perhaps surprising is how well the CS parameters perform given that the fit to the 'exact' data does not capture the maximum or the overall shape of the function. This leaves little guidance on how best to use the CS $H(\beta, W)$ function to fit to a wavefunction that is capable of capturing the long-range behaviour of the electron density, the purpose of this paper.

### 3.2.3. Analysis of the Colle–Salvetti fit parameters

Therefore, further analyses of the CS parameters is performed to give some insight into why they perform so well given the very poor fit to the 'exact' data and to guide us in deriving parameters by

**Table 4.** Comparison of the exact correlation energies, $E_{corr}^{exact}$, with those predicted using the CS formula (2.12) and LYP formula (2.14) using the original CS parameters. The percentage error between exact and calculated values is provided in brackets.

| system | $E_{corr}^{exact}$ (a.u.) | $E_{corr}^{CS}$ (a.u.) | $E_{corr}^{LYP}$ (a.u.) |
|---|---|---|---|
| H⁻ | −0.039 821 282 | −0.030 724 256 (−22.84%) | −0.030 982 748 (−22.19%) |
| Li⁻ | −0.072 6 | −0.070 081 580 (−3.47%) | −0.072 994 582 (+0.54%) |
| B⁻ | −0.145 008 | −0.136 152 443 (−6.11%) | −0.137 130 812 (−5.43%) |
| C⁻ | −0.182 59 | −0.167 977 907 (−8.00%) | −0.170 953 647 (−6.37%) |
| N⁻ | −0.269 813 57 | −0.237 865 345 (−11.84%) | −0.240 434 787 (−10.89%) |
| O⁻ | −0.331 254 1 | −0.299 123 427 (−9.70%) | −0.302 235 871 (−8.76%) |
| F⁻ | −0.399 53 | −0.354 303 731 (−11.32%) | −0.360 627 043 (−9.74%) |
| Na⁻ | −0.419 574 | −0.418 121 783 (−0.35%) | −0.426 850 095 (+1.73%) |
| Al⁻ | −0.483 735 | −0.494 133 170 (+2.15%) | −0.501 806 710 (+3.74%) |
| Si⁻ | −0.520 340 | −0.530 491 961 (+1.95%) | −0.538 604 973 (+3.51%) |
| P⁻ | −0.587 126 | −0.598 669 129 (+1.97%) | −0.606 038 505 (+3.22%) |
| S⁻ | −0.647 570 | −0.659 759 423 (+1.88%) | −0.666 689 428 (+2.95%) |
| Cl⁻ | −0.704 075 | −0.715 798 556 (+1.67%) | −0.723 934 307 (+2.82%) |
| average % error | | 6.40 | 6.29 |

fitting to an anion, discussed in the next section. The importance of the parameters is indicated by the number of significant figures reported by CS [3], i.e. $a > b > c > d$. Handy & Cohen [22] have reported that $a$ is vital in calculating the correlation energy accurately and that (2.9) appears to hold equally well if $0.7 \leq d \leq 0.9$. The denominator $1 + d/\beta$ in $H(\beta, W)$ comes from the form of $\Phi$, (2.5), which governs the extent of the correlation hole [38].

To better understand the significance of each parameter, the sensitivity of the electron correlation energy to small perturbations in each of the $a$, $b$, $c$, $d$ parameters is tested for three systems, He, H⁻ and Na⁻; helium as it is the foundation of the CS method, hydride ion because it produces the largest error in electron correlation energy and the sodium ion as it produces the smallest error in electron correlation energy. Each parameter is varied in the range

$$x - \frac{x}{10} \leq x \leq x + \frac{x}{10}, \quad \text{where } x \in \{a, b, c, d\},$$

whilst the other three are kept fixed at their CS value. The electron correlation energy, $E_{corr}^{CS}$ is calculated using (2.12) for 100 parameter values in this range for each parameter. The high accuracy 25-term Laguerre-based HF wavefunction is used in the case of the helium atom and hydride ion (see §3.1) and the wavefunction from Koga *et al.* [29] is used for the sodium ion.

**The helium atom.** Figure 4 shows four sub-plots, each presenting the electron correlation energy, $E_{corr}^{CS}$ versus a single parameter varied for the neutral helium atom. The red dashed line corresponds to the exact electron correlation energy, $E_{corr}^{exact} = -0.042044381422$ a.u. [7] for helium, and the blue dotted line corresponds to the CS value for the parameter being varied. The gradient of the orange line, the calculated $E_{corr}^{CS}$ values using equation (2.12), represents the sensitivity of $E_{corr}^{CS}$ to each parameter. The case where $b$, $c$, $d$ are fixed whilst $a$ is varied has the steepest gradient showing a small change in $a$ leads to the greatest change in $E_{corr}^{CS}$ compared with the other parameters.

**The hydride ion.** Table 4 shows that applying the CS methodology to the hydride ion produces a very inaccurate electron correlation energy. Figure 5 shows that none of the $a$, $b$, $c$, $d$ parameters produce results close to $E_{corr}^{exact}$ when considering parameter values near those derived by CS; showing that the CS method cannot describe such a weakly bound anion. Electron correlation effects are more dominant in the hydride ion compared with the helium atom owing to a smaller nuclear charge but with equal number of electrons.

**The sodium ion.** In contrast to the hydride ion, table 4 shows that applying the CS methodology to the sodium ion produces a small error in electron correlation energy of approximately 0.35%. Figure 6 shows that the CS value of $a$ almost exactly coincides with the optimum value of $a$ to produce the exact electron correlation energy, which is also true for the $d$ parameter. In the case of the sodium anion, the CS

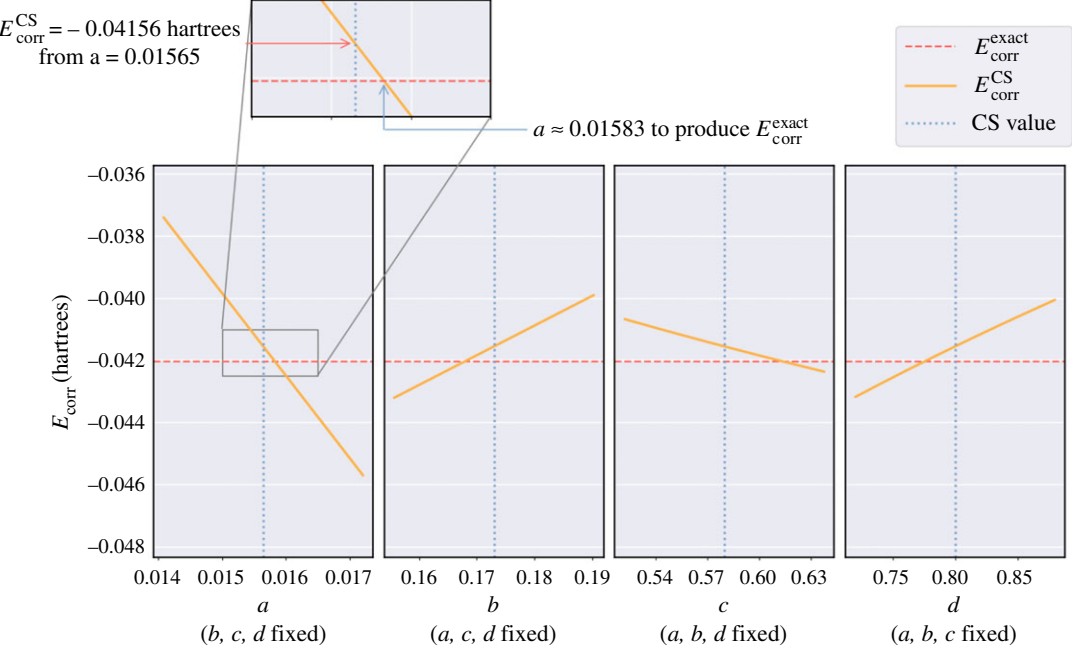

**Figure 4.** Four plots each containing 100 values of the electron correlation energy, $E_{corr}^{CS}$ (orange line) calculated using equation (2.12) for the helium atom versus the parameter values $a$, $b$, $c$, $d$, which are individually varied whilst keeping the other three fixed to their CS value. The red dashed line represents the exact electron correlation energy, $E_{corr}^{exact} = -0.042044381422$ a.u. [7]. The blue dotted line in each plot represents the CS value for the respective parameter: $a = 0.01565$, $b = 0.173$, $c = 0.58$, $d = 0.8$.

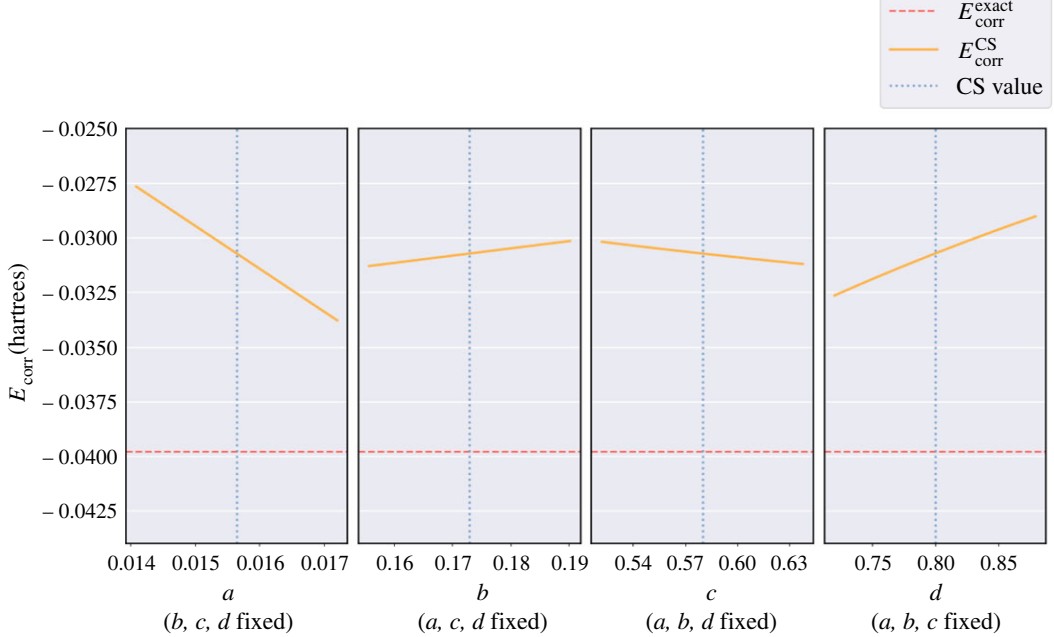

**Figure 5.** Four plots each containing 100 values of the electron correlation energy, $E_{corr}^{CS}$ (orange line) calculated using equation (2.12) for the hydride ion versus the parameter values $a$, $b$, $c$, $d$, which are individually varied whilst keeping the other three fixed to their CS value. The red dashed line represents the exact electron correlation energy, $E_{corr}^{exact} = -0.039821282$ a.u. [7]. The blue dotted line in each plot represents the CS value for the respective parameter: $a = 0.01565$, $b = 0.173$, $c = 0.58$, $d = 0.8$.

values of $b$ and $c$ do not coincide with their optimum values, but the shallow gradient of the orange line in each case shows that the two parameters have little impact on the value of $E_{corr}^{CS}$. This analysis demonstrates that the $a$ parameter is the main control for the accuracy of $E_{corr}^{CS}$. Therefore, for any system where the orange, red and blue lines coincide for $a$, the electron correlation energy will probably be of good accuracy. This provides an explanation for the excellent agreement with the exact correlation energy for Na⁻ in table 4.

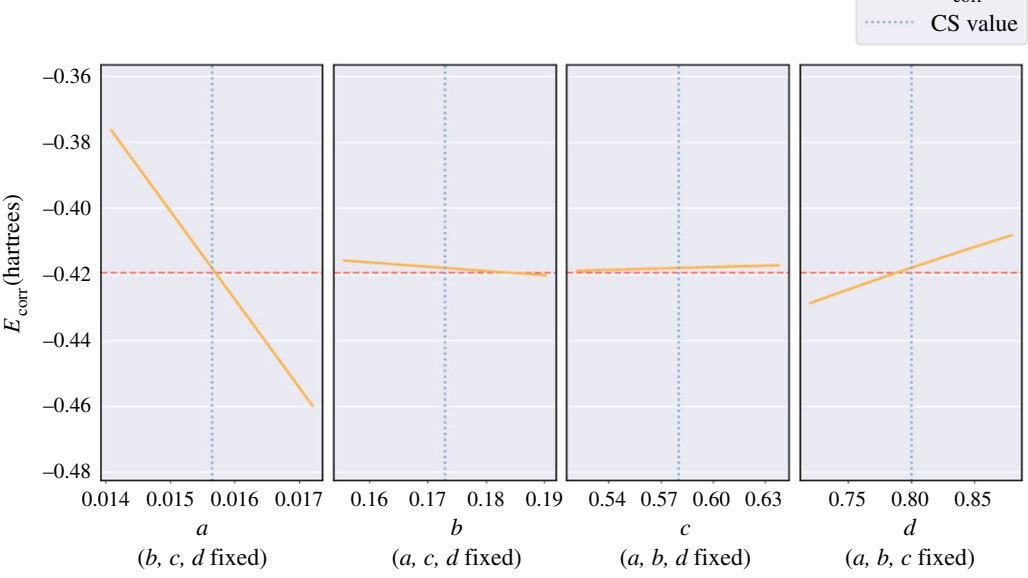

**Figure 6.** Four plots each containing 100 values of the electron correlation energy, $E_{corr}^{CS}$ (orange line) calculated using equation (2.12) for the sodium ion versus the parameter values $a$, $b$, $c$, $d$, which are individually varied while keeping the other three fixed to their CS value. The red dashed line represents the exact electron correlation energy, $E_{corr}^{exact} = -0.419574$ a.u. The blue dotted line in each plot represents the CS value for the respective parameter: $a = 0.01565$, $b = 0.173$, $c = 0.58$, $d = 0.8$.

Due to the empirical nature of the $\{a, b, c, d\}$ parameters in the CS methodology, there is little predictive power as to which systems the CS methodology will work well for. It is clear that the accuracy of the $a$ parameter followed by the $d$ parameter is important for accuracy, with the $b$ and $c$ parameters fluctuating in importance from system to system. The CS method fits to data calculated using the helium atom so one might not expect it to accurately calculate $E_{corr}^{CS}$ for the hydride ion; but it *is* able to accurately calculate $E_{corr}^{CS}$ for the sodium ion. This provides motivation to explore the CS function fit in greater detail, by attempting to capture the physics of weaker bound systems such as the hydride ion.

## 3.3. Parametrizing the Colle–Salvetti formula using an anion

### 3.3.1. Fitting to hydride ion data

The value of $q = 2.29$ valid for the helium atom, related to the width of the Coulomb hole, needs to be rederived for the hydride ion which has a more diffuse Coulomb hole [11]. This is accomplished here by minimizing the square difference, $\Delta$, between the approximation to $E_{corr}$ from equation (2.7) and the exact value of $E_{corr}$, with respect to the $q$ parameter, i.e.

$$\min_{q} \Delta = \min_{q} (E_{corr} - E_{corr}^{exact})^2. \tag{3.1}$$

Equation (2.7) is integrated over all three coordinates $\{r, R, \theta\}$ not using the $H(\beta, W)$ approximation resulting from the second-order Taylor expansion. This technique was verified to work by applying it first to the helium atom where a value of $q = 2.2938$ was calculated, replicating the truncated value of $q = 2.29$ derived by CS. The exact correlation energy for the hydride ion is $E_{corr}^{exact} = -0.039821282$ a.u. [7]; and applying this minimization process results in a value of $q = 1.9398$; this $q$ value is used in the derivation of the parameters $\{a, b, c, d\}$ using the hydride ion where guess value of $a = b = c = d = 0.1$ were used.

The same fitting procedure outlined in the previous section is applied, this time using the high accuracy 25-term Laguerre-based HF orbital for the hydride ion. Table 5 shows the results of fitting the function $H(\beta, W)$, equation (2.8), to the numerically integrated data from (2.15) using the Laguerre-based wavefunction for the hydride ion, over a range of $R$ values with 100 data points within each range. However, in this case the $R$ range is extended to 10 a.u. given that the electron density and Coulomb hole of the hydride ion has a greater radial extent than that of the helium atom [11].

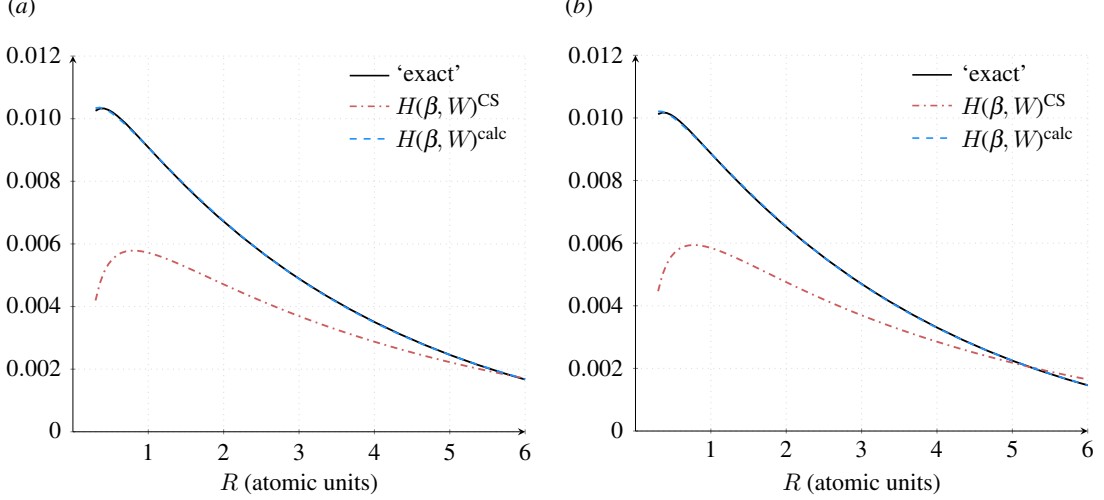

**Figure 7.** Plot of the 'exact' values (2.15) used to determine the parameters $\{a, b, c, d\}$ in the function $H(\beta, W)$ for (a) the hydride anion, $H^-$ and (b) the HF critical nuclear charge system, $Z_C^{HF}$. $H(\beta, W)^{CS}$ uses the parameters determined by CS and $H(\beta, W)^{calc}$ uses the best parameters derived by fitting to (a) $H^-$: $a = 0.020642$, $b = 0.025701$, $c = 0.026314$, $d = 0.790765$ and (b) $Z_C^{HF}$: $a = 0.019539$, $b = 0.023752$, $c = 0.017616$, $d = 0.782764$.

**Table 5.** The value of the correlation energy $E_{corr}^{calc}$ calculated from (2.12) using $\{a, b, c, d\}$ values obtained by using the hydride ion wavefunction and fitting $H(\beta, W)$ to 100 'exact' data points obtained for $R_i \leq R \leq R_f$. The $\chi_{calc}^2$ provides an indication of the fit quality.

| $R_i$ (a.u.) | $R_f$ (a.u.) | a | b | c | d | $E_{corr}^{calc}$ (a.u.) | $\chi_{calc}^2$ |
|---|---|---|---|---|---|---|---|
| 0.10 | 2 | 0.020 345 | 0.024 225 | $1.03 \times 10^{-9}$ | 0.773 917 | **−0.039** 610 141 | 0.000 044 |
| 0.10 | 3 | 0.020 250 | 0.023 784 | $1.57 \times 10^{-8}$ | 0.769 604 | **−0.039** 633 479 | 0.000 034 |
| 0.10 | 4 | 0.020 253 | 0.023 733 | 0.014 994 | 0.773 139 | **−0.039** 774 132 | 0.000 028 |
| 0.10 | 5 | 0.020 202 | 0.023 393 | 0.013 719 | 0.771 200 | **−0.039** 772 169 | 0.000 024 |
| 0.10 | 6 | 0.020 172 | 0.023 119 | 0.012 896 | 0.770 470 | **−0.039** 769 285 | 0.000 021 |
| 0.10 | 10 | 0.020 586 | 0.023 603 | 0.025 307 | 0.799 304 | **−0.039** 776 868 | 0.000 102 |
| 0.30 | 2 | 0.024 402 | 0.054 884 | 0.638 738 | 1.075 912 | **−0.04**0 194 063 | 0.000 001 |
| 0.30 | 3 | 0.024 464 | 0.072 650 | 1.013 840 | 1.107 619 | **−0.04**0 106 871 | 0.000 001 |
| 0.30 | 4 | 0.021 601 | 0.030 635 | 0.089 010 | 0.847 394 | **−0.039** 979 214 | 0.000 003 |
| 0.30 | 5 | 0.020 930 | 0.027 246 | 0.041 588 | 0.806 396 | **−0.039 8**67 507 | 0.000 005 |
| 0.30 | 6 | 0.020 642 | 0.025 701 | 0.026 314 | 0.790 765 | **−0.039 82**8 093 | 0.000 005 |
| 0.30 | 10 | 0.020 847 | 0.027 846 | 0.032 291 | 0.795 230 | **−0.039 83**2 067 | 0.000 016 |
| exact: [7] | | | | | | −0.039 821 282 | |

All function fits tabulated in table 5 result in a suitably accurate calculation of the electron correlation energy for the hydride ion, compared against the literature value. The quality of fit is assessed via $\chi^2$ tests which are all close to zero, signifying a good fit across all ranges of $R$. In general, the $a$ and $d$ parameters converge to similar values, i.e. $a \approx 0.020$ and $d \approx 0.8$, regardless of the $R$ range. Interestingly, by fitting to the hydride ion and using a different $q$ value, $a$ and $d$ are similar to the values derived by CS for helium [3]. As shown in §3.2.3 the two dominant parameters in the CS method are $a$ and $d$ so the newly derived values appear to be following a similar trend.

As perhaps expected, slightly more accurate hydride ion correlation energies were found when considering a larger $R_f$ value, see $0.3 \leq R \leq 6$ a.u. in table 5. Although like the helium atom any chosen $R$-range results in a suitably accurate fit to the 'exact' data used in the fit and the exact electron correlation energy.

Figure 7a overlays $H(\beta, W)^{calc}$, calculated using the $\{a, b, c, d\}$ values from fitting in the range $0.3 \leq R \leq 6$ a.u. with the numerically integrated data points and the CS form of $H(\beta, W)$. As in figure 3,

**Table 6.** Comparison of the exact correlation energies, $E_{corr}^{exact}$, with those predicted using the CS formula (2.12) and LYP formula (2.14) using the parameters derived from fitting to the hydride ion. The percentage error between exact and calculated values is provided in brackets.

| system | $E_{corr}^{exact}$ (a.u.) | $E_{corr}^{CS}$ (a.u.) | $E_{corr}^{LYP}$ (a.u.) |
|---|---|---|---|
| $H^-$ | −0.039 821 282 | −0.039 827 668 (+0.02%) | −0.040 562 809 (−1.86%) |
| $Li^-$ | −0.072 6 | −0.104 022 825 (+43.28%) | −0.105 305 996 (+45.05%) |
| $B^-$ | −0.145 008 | −0.179 012 169 (+23.45%) | −0.179 294 619 (+23.64%) |
| $C^-$ | −0.182 59 | −0.210 804 315 (+15.45%) | −0.212 266 647 (+16.25%) |
| $N^-$ | −0.269 813 57 | −0.293 375 647 (+8.73%) | −0.294 773 434 (+9.25%) |
| $O^-$ | −0.331 254 1 | −0.366 995 910 (+10.79%) | −0.368 608 769 (+11.28%) |
| $F^-$ | −0.399 53 | −0.433 188 820 (+8.42%) | −0.435 995 193 (+9.13%) |
| $Na^-$ | −0.419 574 | −0.510 131 323 (+21.58%) | −0.513 515 741 (+22.39%) |
| $Al^-$ | −0.483 735 | −0.591 212 608 (+22.22%) | −0.593 552 033 (+22.70%) |
| $Si^-$ | −0.520 340 | −0.627 150 705 (+20.53%) | −0.630 054 054 (+21.09%) |
| $P^-$ | −0.587 126 | −0.703 534 620 (+19.83%) | −0.706 360 508 (+20.31%) |
| $S^-$ | −0.647 570 | −0.772 555 322 (+19.30%) | −0.775 344 271 (+19.73%) |
| $Cl^-$ | −0.704 075 | −0.835 999 674 (+18.74%) | −0.839 499 544 (+19.23%) |
| average % error | | 17.87 | 18.61 |

figure 7a shows that $H(\beta, W)^{CS}$ is unable to correctly describe the peak height of the numerical data even though it now has the correct shape.

Table 6 shows calculated anionic electron correlation energies from equation (2.12) using the parameter values $a = 0.020642$, $b = 0.025701$, $c = 0.026314$, $d = 0.790765$ derived in table 5 for the range $0.3 \leq R \leq 6$ a.u. which produced the closest electron correlation energy value to the literature value. Table 6 shows that accurately fitting to data generated using a hydride ion orbital results in a function form which poorly approximates the electron correlation energy for general anions. It performs only marginally better than the accurate function fit to helium data, and does not improve on the original CS formulation, seen in table 4.

It was shown in §3.2 that an accurate function fit to helium data results in a form of $H(\beta, W)$ too specific to helium-like systems, lacking the versatility to describe other chemical systems. Again this is probably the case when fitting to the hydride ion where figure 7a shows an almost perfect overlap between the numerical data points and the calculated fitted function. The non-transferability of the optimized parameters $\{a, b, c, d\}$ to other anions perhaps demonstrates the non-physical nature of the parameters highlighted by Tsuneda et al. [39]. Nevertheless, the shape of the original $H(\beta, W)^{CS}$ has correctly adapted to the hydride system highlighting its versatility.

### 3.3.2. Fitting to critical nuclear charge data

This brings into question the physical justification of fitting to data generated using an RHF hydride wavefunction. A more suitable candidate might be the system which has the minimum nuclear charge required to bind two electrons using RHF; the critical nuclear charge system, $Z_C^{HF} = 1.031177528$. The 25-term Laguerre-based wavefunction calculated using RHF for a two-electron atom with nuclear charge 1.031177528 was used to generate the 'exact' data and the fit to $H(\beta, W)$. As with the hydride ion, the $q$ value was recalculated using this $Z_C^{HF}$ wavefunction resulting in a value of $q = 1.9672$. The result of fitting the function $H(\beta, W)$, equation (2.8), to the numerically integrated data from equation (2.15) using the Laguerre-based wavefunction for the $Z_C^{HF}$ system, over a range of $R$ values with 100 data points within each range, provided results qualitatively similar to those in table 5.

All function fits provide a good correlation energy, when compared with the exact literature value [7], and the best-fit results from the range $0.30 \leq R \leq 6$ a.u. (see electronic supplementary material) with $a = 0.019539$, $b = 0.023752$, $c = 0.017616$, $d = 0.782764$. Figure 7b shows that the form of $H(\beta, W)$ is also very similar to the hydride ion results.

**Table 7.** Comparison of the exact correlation energies, $E_{corr}^{exact}$, with those predicted using the CS formula (2.12) and LYP formula (2.14) using the parameters derived from fitting to the $Z_C^{HF}$ system. The percentage error between exact and calculated values is provided in brackets.

| system | $E_{corr}^{exact}$ (a.u.) | $E_{corr}^{CS}$ (a.u.) | $E_{corr}^{LYP}$ (a.u.) |
|---|---|---|---|
| H$^-$ | −0.039 821 282 | −0.038 541 720 (−3.21%) | −0.039 220 793 (−1.51%) |
| $Z_c^{HF}$ | −0.039 715 117 4 | −0.039 710 035 (−0.01%) | −0.039 709 842 (−0.01%) |
| Li$^-$ | −0.072 6 | −0.100 014 173 (+37.76%) | −0.101 153 655 (+39.33%) |
| B$^-$ | −0.145 008 | −0.171 045 086 (+17.96%) | −0.171 286 614 (+18.12%) |
| C$^-$ | −0.182 59 | −0.200 929 370 (+10.04%) | −0.202 206 908 (+10.74%) |
| N$^-$ | −0.269 813 57 | −0.279 656 798 (+3.65%) | −0.280 889 693 (+4.11%) |
| O$^-$ | −0.331 254 1 | −0.349 702 900 (+5.57%) | −0.351 124 198 (+5.99%) |
| F$^-$ | −0.399 53 | −0.412 517 095 (+3.25%) | −0.414 972 367 (+3.87%) |
| Na$^-$ | −0.419 574 | −0.485 175 319 (+15.64%) | −0.488 129 631 (+16.34%) |
| Al$^-$ | −0.483 735 | −0.561 535 888 (+16.08%) | −0.563 540 964 (+16.49%) |
| Si$^-$ | −0.520 340 | −0.595 242 961 (+14.39%) | −0.597 751 607 (+14.88%) |
| P$^-$ | −0.587 126 | −0.667 983 934 (+13.77%) | −0.670 445 763 (+14.19%) |
| S$^-$ | −0.647 570 | −0.733 594 964 (+13.28%) | −0.736 032 085 (+13.66%) |
| Cl$^-$ | −0.704 075 | −0.793 765 379 (+12.74%) | −0.796 828 717 (+13.17%) |
| average % error | | 11.95 | 12.32 |

Table 7 shows that the agreement in calculated electron correlation energies with the exact values still lacks adequate accuracy, but provides an improvement when compared with the accurate fits to the helium atom, table 3, and the hydride ion, table 6.

# 4. Conclusion

We have investigated the CS methodology based on their 1975 paper [3] and how the CS and LYP functionals perform when applied to atomic anions. The sensitivity in fitting parameters $\{a, b, c, d\}$ was visually studied for the hydride, helium and sodium ions. Next, a hydride ion orbital and an orbital calculated using the critical nuclear charge for RHF theory, $Z_C^{HF} = 1.031177528$ [7], were used in place of the helium orbital to elucidate whether or not the resulting form of $H(\beta, W)$ better captures the physics of anions. The key conclusions are as follows:

1. In general, both the CS and LYP functionals offer acceptable accuracy (errors less than 12%) for the anions considered, and the accuracy improves substantially as the nuclear charge increases. However, they are unable to accurately calculate the electron correlation energy for the smaller, more weakly bound hydride ion.
2. Calculating a more accurate function fit of $H(\beta, W)$ to the neutral helium atom using (i) the Clementi [4] orbital used by CS [3] and (ii) a very accurate HF wavefunction results in a more accurate electron correlation energy for the helium atom but is detrimental to the overall CS method when applied to other anions resulting in large errors for calculated electron correlation energies. The loss of accuracy is attributed to the tightness of the fit; relaxing the fit provides results similar to those of CS with only a marginal improvement for anionic systems.
3. It is shown that the $H(\beta, W)$ derived by CS is inaccurate when compared with the calculated helium data it is designed to describe when $R > 2$ a.u. This poses an interesting conundrum as for the CS mechanism to work it requires a loose fit to the data, which is not quantifiable.
4. Using a high accuracy HF wavefunction for the hydride ion or the $Z_C^{HF}$ system to determine the densities used in the fitting and $q$, a parameter related to the mass of electrons in the correlation hole, chosen to reproduce the anion correlation energy, resulted in a very accurate electron correlation energy for the anion (0.02% difference from the exact value for the hydride ion and 0.01% difference for the $Z_C^{HF}$ system), but like with the helium atom reparametrization, it results in a general form which does not contain sufficient flexibility to accurately model other systems.

Overall, we have found that using weakly bound anionic orbitals in place of the neutral helium atom within the CS methodology does not produce a more accurate description of anions. It is possible to manipulate the parameters in the function $H(\beta, W)$ manually for slight improvement, but this  offers no physically motivated foundation on which to build a correlation functional for use in approximate DFT. Although not improving on the CS parameters, the parameters $\{a, b, c, d\}$ derived from the critical nuclear charge system were shown to outperform the accurate helium and hydride fittings  in calculating the electron correlation energies of anions considered; suggesting it may offer a firm foundation for functional development which goes beyond the CS methodology; this is currently underway.

Data accessibility. The coefficients of the 25-term Laguerre-based wavefunctions are provided in the electronic supplementary material along with a table assessing the fit quality of the correlation energy obtained with parameters derived using the $Z_C^{HF}$ wavefunctions.

Authors' contributions. A.L.B. developed and implemented the programs used to generate the data in the paper. M.T. and A.L.B. produced the data and all authors contributed to the data analysis. The manuscript was written by A.L.B. and H.C., and the study was conceived and devised by H.C. All authors approved the final version of the manuscript.

Competing interests. We declare we have no competing interests.

Funding. This work is supported by EPSRC, grant no. EP/R011265/1.

Acknowledgements. H.C. and A.L.B. would like to thank EPSRC for financial support (grant no. EP/R011265/1). M.T. would like to thank the Tertiary Education Trust Fund (TETFund) of Nigeria for PhD sponsorship through the Benue State University, Makurdi.

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
