## [Peer Review File · Royal Society Open Science]

Review History

RSOS-211333.R0 (Original submission)

Review form: Reviewer 1

Is the manuscript scientifically sound in its present form?

Yes

Are the interpretations and conclusions justified by the results?

Yes

Is the language acceptable?

Yes

Do you have any ethical concerns with this paper?

No

Have you any concerns about statistical analyses in this paper?

No

Recommendation?

Accept with minor revision (please list in comments)

Comments to the Author(s)

The manuscript by Cox and coworkers revisits the Colle-Salvetti (CS) formula, which underlies the LYP correlation functional and is thus a cornerstone of density-functional theory. In particular, it investigates whether a re-parametrization of the CS formula (which dates back to 1975 and is based on a Hartree-Fock wavefunction obtained by Clementi in 1965) using highly accurate data available today could be beneficial. It then extends this by performing a re-parametrization using anions, and assesses whether this improves the description of electron correlation in anions.

Even though the results are mainly negative (i.e., a re-parametrization does hardly improve compared to the original CS formula), they are of great importance for understanding the success of the LYP functional and to provide guidance towards improving correlation functionals. The authors' analysis is instructive, and discusses these insights very clearly. I particularly like the sensitivity analysis of different parameters in the CS formula / LYP functional.

I only have two minor comments that the authors might address in a revised version of their manuscript:

1) The authors refer to "overfitting" several times when discussing their observation that the accuracy of the CS formula is worse if a fit that is more accurate at larger distances is used. I was a bit confused by this, as I think that "overfitting" usually refers to a case where a very flexible function is fitted to a small data set, resulting in spurious oscillations of the fit. This is not the case here, and the problem is not the fitting procedure, but the underlying data, i.e., the fact that the helium atom might be a poor model. Maybe the authors can find a better term here, or at least add a short explanation of their use of the term "overfitting".

2) I think it might be useful to also include "Fit 2" in Figure 3.

Review form: Reviewer 2

Is the manuscript scientifically sound in its present form?

Yes

Are the interpretations and conclusions justified by the results?

Yes

Is the language acceptable?

Yes

Do you have any ethical concerns with this paper?

No

Have you any concerns about statistical analyses in this paper?

No

Recommendation?

Major revision is needed (please make suggestions in comments)

Comments to the Author(s)

In this article, the authors revisit the fitting of parameters in the Colle-Salvetti (CS) formula. In particular, they replace the original Hartree-Fock (HF) wave functions used by Colle and Salvetti with high-accuracy HF wave functions computed using an approach developed in their group. These high-accuracy wave functions correctly capture the electron-nuclear cusp while remaining the electron-electron correlation energy is described using the CS formula. They then investigate various refitting of the CS equations for neutral, cationic, and anionic nuclei. Their primary results suggest:

- (i) Refitting the CS formula with high-accuracy HF wave functions can improve the description of the atom to which it was fitted, but is not generally transferable to other atoms;
- (ii) The original CS formula performs badly for anionic systems, particularly the hydride anion;
- (iii) Refitting the CS formula to a high-accuracy HF wave function for hydride dramatically improves the correlation energy for H^{-} , but is not transferable to other anions.

Given the significance of the CS formula to density functional approximations such as the LYP functional, revisiting the original CS parameterisation using high-accuracy wave functions for these more exotic systems is timely and novel. However, there are areas of the manuscript that I believe require some further clarification. I would therefore support publication once the reviewers have addressed the comments below:

- (1) The abstract does not read clearly. In particular, the third sentence starting "It is shown that..." is very unclear and I would suggest the authors consider rewriting it.
- (2) Equation 2.15 seems to be missing a "dr" for the first integral (0 to inf)
- (3) In Section 3b(ii), it concerns me that the authors appear to have found different sets of CS parameters for the same R_i and R_f values using their optimisation method or by starting from the original CS parameters. (e.g. in Table 2 they get $a=0.011485$, $b=0.024418$, $c\sim 0$, $d=0.664135$, but in the paragraphs below they describe 'Fit 2' with $a=0.01628$, $b=0.18438$, $c=0.57594$, $d=0.80562$.) This suggests that the parameter optimisation is starting-point dependent. Can the authors clarify what starting point they use, and can they confirm whether the global best fit has been identified? Understanding the choice of starting point is essential for reproducibility. How would "Fit 1" and "Fit 2" compare to the graphical representation in Figure 3?
- (4) The failure of CS for the hydride anion is spectacular, and the variable performance across other anions is rightly highlighted as a point of concern. What is even more surprising is that fitting the CS parameters to the hydride anion does not lead to a transferable approach. The authors describe this as an "over-fitting" issue, but that seems unlikely to account for such a large error. I wonder if the authors can provide more physical insight into this failure? To me, it looks like the failure for anionic systems is a static correlation effect, which can be inferred from the presence of UHF symmetry breaking. In particular, previous work from the authors [doi:10.1098/rsos.181357] has shown that the HF approximation predicts a large error in the one-particle radial distance $\langle r_1 \rangle$ for hydride ($\langle r_1 \rangle = 2.71$ in fully-correlated vs $\langle r_1 \rangle = 2.5$ in HF). If the RHF density is a poor initial approximation, then the CS formula must perform the dual task of relaxing the one-particle energy (by relaxing the orbitals) and capturing the two-electron correlation. This may explain why the hydride-fitted parameters overestimate the correlation energy in the heavier cations, where perhaps the one-particle relaxation is less important. A simple test would be generate the 2nd-order HF density for hydride using the exact one-particle density obtained from the fully-correlated wave function in [doi:10.1098/rsos.181357] and use this to fit the CS parameters. This should leave only electron-electron correlation errors.

(5) The authors should be careful about their choice of the HF critical nuclear charge in Section 3c(ii). They choose the charge $Z_c = 1.031177528$ which they have previously identified as the point where the RHF energy is degenerate with the ionised system. However, other work including [doi:10.1063/1.4871018], [doi:10.1103/PhysRevA.101.062504], and [doi:10.1063/5.0043105] define the critical nuclear charge as the point where the occupied orbital energy becomes zero, corresponding to $Z_c=0.828161008$. This alternative definition might be considered more physical as it is the point where the electrons suddenly start to detach from the nucleus. Furthermore, the asymptotic behaviour of the radial density becomes $\exp(-a \sqrt{r})$, in contrast to the standard $\exp(-a r)$. Given these alternative definitions for the nuclear charge, and the potential importance of the long-range density behaviour in fitting the CS formula, the authors should provide a better justification for their choice of Z_c . In their current results, it is not surprising that the Z_c and H- fitted parameters give such similar results as the RHF wave functions are qualitatively very similar at these two charges, i.e. the electrons are still bound at $Z_c=1.031177528$. It would therefore be interesting to compare the optimised CS parameters at both definitions $Z_c=1.031177528$ and $Z_c=0.828161008$ to understand the significance of the critical binding threshold.

(6) Having identified the challenges of fitting the CS parameters, can the authors comment on potential ways to overcome these challenges?

(7) In Supporting Information Table IV, the caption refers to "The percentage error between exact and calculated values is provided in brackets", but I cannot find any brackets in the table.

Decision letter (RSOS-211333.R0)

Dear Dr Cox:

Title: Reparametrization of the Colle-Salvetti Formula
Manuscript ID: RSOS-211333

The editor assigned to your manuscript has now received comments from reviewers. We would like you to revise your paper in accordance with the referee and Subject Editor suggestions which can be found below (not including confidential reports to the Editor). Please note this decision does not guarantee eventual acceptance.

Please submit your revised paper before 05-Nov-2021. Please note that the revision deadline will expire at 00.00am on this date. If we do not hear from you within this time then it will be assumed that the paper has been withdrawn. In exceptional circumstances, extensions may be possible if agreed with the Editorial Office in advance. We do not allow multiple rounds of revision so we urge you to make every effort to fully address all of the comments at this stage. If deemed necessary by the Editors, your manuscript will be sent back to one or more of the original reviewers for assessment. If the original reviewers are not available we may invite new reviewers.

Yours sincerely,
Dr Ellis Wilde
Publishing Editor, Journals

On behalf of the Subject Editor Professor Anthony Stace and the Associate Editor Dr Annette Trunschke.

RSC Associate Editor

Comments to the Author:

Two basically positive reviews were received for the work. The reviewers provide valuable suggestions for improvement that should be taken into account by the authors when preparing a revised version.

RSC Subject Editor

Comments to the Author:

(There are no comments.)

Reviewers' Comments to Author:

Reviewer: 1

Comments to the Author(s)

The manuscript by Cox and coworkers revisits the Colle-Salvetti (CS) formula, which underlies the LYP correlation functional and is thus a cornerstone of density-functional theory. In particular, it investigates whether a re-parametrization of the CS formula (which dates back to 1975 and is based on a Hartree-Fock wavefunction obtained by Clementi in 1965) using highly accurate data available today could be beneficial. It then extends this by performing a re-parametrization using anions, and assesses whether this improves the description of electron correlation in anions.

Even though the results are mainly negative (i.e., a re-parametrization does hardly improve compared to the original CS formula), they are of great importance for understanding the success of the LYP functional and to provide guidance towards improving correlation functionals. The authors' analysis is instructive, and discusses these insights very clearly. I particularly like the sensitivity analysis of different parameters in the CS formula / LYP functional.

I only have two minor comments that the authors might address in a revised version of their manuscript:

1) The authors refer to "overfitting" several times when discussing their observation that the accuracy of the CS formula is worse if a fit that is more accurate at larger distances is used. I was a bit confused by this, as I think that "overfitting" usually refers to a case where a very flexible function is fitted to a small data set, resulting in spurious oscillations of the fit. This is not the case here, and the problem is not the fitting procedure, but the underlying data, i.e., the fact that the helium atom might be a poor model. Maybe the authors can find a better term here, or at least add a short explanation of their use of the term "overfitting".

2) I think it might be useful to also include "Fit 2" in Figure 3.

Reviewer: 2

Comments to the Author(s)

In this article, the authors revisit the fitting of parameters in the Colle-Salvetti (CS) formula. In particular, they replace the original Hartree-Fock (HF) wave functions used by Colle and Salvetti with high-accuracy HF wave functions computed using an approach developed in their group. These high-accuracy wave functions correctly capture the electron-nuclear cusp while remaining the electron-electron correlation energy is described using the CS formula. They then investigate various refitting of the CS equations for neutral, cationic, and anionic nuclei. Their primary results suggest:

(i) Refitting the CS formula with high-accuracy HF wave functions can improve the description of the atom to which it was fitted, but is not generally transferable to other atoms;

(ii) The original CS formula performs badly for anionic systems, particularly the hydride anion;

(iii) Refitting the CS formula to a high-accuracy HF wave function for hydride dramatically improves the correlation energy for H^{-} , but is not transferable to other anions.

Given the significance of the CS formula to density functional approximations such as the LYP functional, revisiting the original CS parameterisation using high-accuracy wave functions for these more exotic systems is timely and novel. However, there are areas of the manuscript that I believe require some further clarification. I would therefore support publication once the reviewers have addressed the comments below:

(1) The abstract does not read clearly. In particular, the third sentence starting "It is shown that..." is very unclear and I would suggest the authors consider rewriting it.

(2) Equation 2.15 seems to be missing a "dr" for the first integral (0 to inf)

(3) In Section 3b(ii), it concerns me that the authors appear to have found different sets of CS parameters for the same R_i and R_f values using their optimisation method or by starting from the original CS parameters. (e.g. in Table 2 they get $a=0.011485$, $b=0.024418$, $c\sim 0$, $d=0.664135$, but in the paragraphs below they describe 'Fit 2' with $a=0.01628$, $b=0.18438$, $c=0.57594$, $d=0.80562$.) This suggests that the parameter optimisation is starting-point dependent. Can the authors clarify what starting point they use, and can they confirm whether the global best fit has been identified? Understanding the choice of starting point is essential for reproducibility. How would "Fit 1" and "Fit 2" compare to the graphical representation in Figure 3?

(4) The failure of CS for the hydride anion is spectacular, and the variable performance across other anions is rightly highlighted as a point of concern. What is even more surprising is that fitting the CS parameters to the hydride anion does not lead to a transferable approach. The authors describe this as an "over-fitting" issue, but that seems unlikely to account for such a large error. I wonder if the authors can provide more physical insight into this failure? To me, it looks like the failure for anionic systems is a static correlation effect, which can be inferred from the presence of UHF symmetry breaking. In particular, previous work from the authors [doi:10.1098/rsos.181357] has shown that the HF approximation predicts a large error in the one-particle radial distance for hydride ($= 2.71$ in fully-correlated vs $= 2.5$ in HF). If the RHF density is a poor initial approximation, then the CS formula must perform the dual task of relaxing the one-particle energy (by relaxing the orbitals) and capturing the two-electron correlation. This may explain why the hydride-fitted parameters overestimate the correlation energy in the heavier cations, where perhaps the one-particle relaxation is less important. A simple test would be generate the 2nd-order HF density for hydride using the exact one-particle density obtained from the fully-correlated wave function in [doi:10.1098/rsos.181357] and use this to fit the CS parameters. This should leave only electron-electron correlation errors.

(5) The authors should be careful about their choice of the HF critical nuclear charge in Section 3c(ii). They choose the charge $Z_c = 1.031177528$ which they have previously identified as the point where the RHF energy is degenerate with the ionised system. However, other work including [doi:10.1063/1.4871018], [doi:10.1103/PhysRevA.101.062504], and [doi:10.1063/5.0043105] define the critical nuclear charge as the point where the occupied orbital energy becomes zero, corresponding to $Z_c=0.828161008$. This alternative definition might be considered more physical as it is the point where the electrons suddenly start to detach from the nucleus. Furthermore, the asymptotic behaviour of the radial density becomes $\exp(-a \sqrt{r})$, in contrast to the standard $\exp(-a r)$. Given these alternative definitions for the nuclear charge, and the potential importance of the long-range density behaviour in fitting the CS formula, the authors should provide a better justification for their choice of Z_c . In their current results, it is not surprising that the Z_c and H- fitted parameters give such similar results as the RHF wave functions are qualitatively very similar at these two charges, i.e. the electrons are still bound at $Z_c=1.031177528$. It would therefore be interesting to compare the optimised CS parameters at both definitions $Z_c=1.031177528$ and $Z_c=0.828161008$ to understand the significance of the critical binding threshold.

(6) Having identified the challenges of fitting the CS parameters, can the authors comment on potential ways to overcome these challenges?

(7) In Supporting Information Table IV, the caption refers to "The percentage error between exact and calculated values is provided in brackets", but I cannot find any brackets in the table.

Author's Response to Decision Letter for (RSOS-211333.R0)

See Appendix A.

RSOS-211333.R0 (Original submission)

Review form: Reviewer 1

Is the manuscript scientifically sound in its present form?

Yes

Are the interpretations and conclusions justified by the results?

Yes

Is the language acceptable?

Yes

Do you have any ethical concerns with this paper?

No

Have you any concerns about statistical analyses in this paper?

No

Recommendation?

Accept as is

Comments to the Author(s)

The authors have addressed all my previous comments.

Review form: Reviewer 2

Is the manuscript scientifically sound in its present form?

Yes

Are the interpretations and conclusions justified by the results?

Yes

Is the language acceptable?

Yes

Do you have any ethical concerns with this paper?

No

Have you any concerns about statistical analyses in this paper?

No

Recommendation?

Accept as is

Comments to the Author(s)

I thank the authors for their excellent responses and I congratulate them on a very interesting study.

Decision letter (RSOS-211333.R1)

Dear Dr Cox:

Title: Reparametrization of the Colle-Salvetti Formula
Manuscript ID: RSOS-211333.R1

It is a pleasure to accept your manuscript in its current form for publication in Royal Society Open Science. The chemistry content of Royal Society Open Science is published in collaboration with the Royal Society of Chemistry.

Yours sincerely,
Dr Ellis Wilde
Publishing Editor, Journals

On behalf of the Subject Editor Professor Anthony Stace and the Associate Editor Dr Annette Trunschke.

RSC Associate Editor
Comments to the Author:
(There are no comments.)

RSC Subject Editor
Comments to the Author:
(There are no comments.)

Reviewer(s)' Comments to Author:

Reviewer: 1

Comments to the Author(s)

The authors have addressed all my previous comments.

Reviewer: 2

Comments to the Author(s)

I thank the authors for their excellent responses and I congratulate them on a very interesting study.

Appendix A

Response to Reviewer Comments: MS Reference Number: RSOS-211333

MS Title: Reparametrization of the Colle-Salvetti Formula
MS Authors: Baskerville, Adam; Targema, Msugh; Cox, Hazel

We thank the referees for their positive and constructive comments. In response, we have made the following changes. The response to the reviewers' comments is below in red/italics and added or revised text is highlighted in blue in the manuscript.

Reviewers' Comments to Author:

Reviewer: 1

Comments to the Author(s)

The manuscript by Cox and coworkers revisits the Colle-Salvetti (CS) formula, which underlies the LYP correlation functional and is thus a cornerstone of density-functional theory. In particular, it investigates whether a re-parametrization of the CS formula (which dates back to 1975 and is based on a Hartree-Fock wavefunction obtained by Clementi in 1965) using highly accurate data available today could be beneficial. It then extends this by performing a re-parametrization using anions, and assesses whether this improves the description of electron correlation in anions.

Even though the results are mainly negative (i.e., a re-parametrization does hardly improve compared to the original CS formula), they are of great importance for understanding the success of the LYP functional and to provide guidance towards improving correlation functionals. The authors' analysis is instructive, and discusses these insights very clearly. I particularly like the sensitivity analysis of different parameters in the CS formula / LYP functional.

I only have two minor comments that the authors might address in a revised version of their manuscript:

1) The authors refer to "overfitting" several times when discussing their observation that the accuracy of the CS formula is worse if a fit that is more accurate at larger distances is used. I was a bit confused by this, as I think that "overfitting" usually refers to a case where a very flexible function is fitted to a small data set, resulting in spurious oscillations of the fit. This is not the case here, and the problem is not the fitting procedure, but the underlying data, i.e., the fact that the helium atom might be a poor model. Maybe the authors can find a better term here, or at least add a short explanation of their use of the term "overfitting".

We have made the following changes in the text to address this:

Page 6 second para "... found that over-fitting could be problematic" replaced by "... found that a tighter fit could be problematic"

Page 11 first para comment deleted

Page 11 second para "by not overfitting" replaced with "by relaxing the fit"

Page 16 mention of overfitting deleted

Page 18 conclusions "The loss of accuracy is attributed to over-fitting;" replaced by "The loss of accuracy is attributed to the tightness of the fit;"

2) I think it might be useful to also include "Fit 2" in Figure 3.

Thank you for this suggestion - Fit 2 has now been included in Figure 3.

Reviewer: 2

Comments to the Author(s)

In this article, the authors revisit the fitting of parameters in the Colle-Salvetti (CS) formula. In particular, they replace the original Hartree-Fock (HF) wave functions used by Colle and Salvetti with high-accuracy HF wave functions computed using an approach developed in their group. These high-accuracy wave functions correctly capture the electron-nuclear cusp while remaining the electron-electron correlation energy is described using the CS formula. They then investigate various refitting of the CS equations for neutral, cationic, and anionic nuclei. Their primary results suggest:

- (i) Refitting the CS formula with high-accuracy HF wave functions can improve the description of the atom to which it was fitted, but is not generally transferable to other atoms;
- (ii) The original CS formula performs badly for anionic systems, particularly the hydride anion;
- (iii) Refitting the CS formula to a high-accuracy HF wave function for hydride dramatically improves the correlation energy for H^{-} , but is not transferable to other anions.

Given the significance of the CS formula to density functional approximations such as the LYP functional, revisiting the original CS parameterisation using high-accuracy wave functions for these more exotic systems is timely and novel. However, there are areas of the manuscript that I believe require some further clarification. I would therefore support publication once the reviewers have addressed the comments below:

- (1) The abstract does not read clearly. In particular, the third sentence starting “It is shown that...” is very unclear and I would suggest the authors consider rewriting it.

Thank you for this comment. We have replaced the sentence beginning “It is shown ...” with the following shorter sentence:

“Fitting to the hydride ion or the two-electron system just prior to electron detachment at the HF level of theory does not, in general, improve the calculated correlation energies using the parameters derived from the CS/LYP method.”

- (2) Equation 2.15 seems to be missing a “dr” for the first integral (0 to inf)

This was a typographic error which has now been corrected.

- (3) In Section 3b(ii), it concerns me that the authors appear to have found different sets of CS parameters for the same R_i and R_f values using their optimisation method or by starting from the original CS parameters. (e.g. in Table 2 they get $a=0.011485$, $b=0.024418$, $c=0$, $d=0.664135$, but in the paragraphs below they describe ‘Fit 2’ with $a=0.01628$, $b=0.18438$, $c=0.57594$, $d=0.80562$.) This suggests that the parameter optimisation is starting-point dependent. Can the authors clarify what starting point they use, and can they confirm whether the global best fit has been identified? Understanding the choice of starting point is essential for reproducibility. How would “Fit 1” and “Fit 2” compare to the graphical representation in Figure 3?

Due to the non-uniqueness of the function fits we are not able to confirm whether or not we have obtained the global minimum. Extensive testing of the fitting program was conducted during development and even though different optimised a, b, c, d values result from different starting points,

all the resulting functions fit the data well. This is mentioned on page 7 (point iv) and we refer readers to the thesis of one of the co-authors.

However, we have now included the starting values used for Fit 1 so that both fits reported include the starting values.

We have also added the following text (page 8): “This dependence of the optimised fit parameters on the starting values is a common problem, see for example N. Chernov, Q. Huang, H. Ma, "Is the Best Fitting Curve Always Unique?", Journal of Mathematics, vol. 2013, Article ID 753981, 5 pages, 2013}”

(4) The failure of CS for the hydride anion is spectacular, and the variable performance across other anions is rightly highlighted as a point of concern. What is even more surprising is that fitting the CS parameters to the hydride anion does not lead to a transferable approach. The authors describe this as an “over-fitting” issue, but that seems unlikely to account for such a large error. I wonder if the authors can provide more physical insight into this failure? To me, it looks like the failure for anionic systems is a static correlation effect, which can be inferred from the presence of UHF symmetry breaking. In particular, previous work from the authors [doi:10.1098/rsos.181357] has shown that the HF approximation predicts a large error in the one-particle radial distance for hydride ($= 2.71$ in fully-correlated vs $= 2.5$ in HF). If the RHF density is a poor initial approximation, then the CS formula must perform the dual task of relaxing the one-particle energy (by relaxing the orbitals) and capturing the two-electron correlation. This may explain why the hydride-fitted parameters overestimate the correlation energy in the heavier cations, where perhaps the one-particle relaxation is less important. A simple test would be generate the 2nd-order HF density for hydride using the exact one-particle density obtained from the fully-correlated wave function in [doi:10.1098/rsos.181357] and use this to fit the CS parameters. This should leave only electron-electron correlation errors.

Thank you for this insightful comment. A contributing factor to the failure of the CS method for the hydride ion may indeed be due to the failure of the anionic systems to model static correlation effects. However, given the non-physical nature of the a, b, c, d parameters (which has also been highlighted in the work of Tsuneda et al.) and numerous approximations in the CS method (see for example reference [5] and [6] in the paper) we did not feel it appropriate to comment on specific physical factors as there are way too many possible factors (for example [6] which also indicates that the success of the CS method is likely due to fortuitous cancellation of errors).

We have added the following text (p16) “The non-transferability of the optimised parameters $\{a, b, c, d\}$ to other anions perhaps demonstrates the non-physical nature of the parameters highlighted by Tsuneda et al.”

(5) The authors should be careful about their choice of the HF critical nuclear charge in Section 3c(ii). They choose the charge $Z_c = 1.031177528$ which they have previously identified as the point where the RHF energy is degenerate with the ionised system. However, other work including [doi:10.1063/1.4871018], [doi:10.1103/PhysRevA.101.062504], and [doi:10.1063/5.0043105] define the critical nuclear charge as the point where the occupied orbital energy becomes zero, corresponding to $Z_c = 0.828161008$. This alternative definition might be considered more physical as it is the point where the electrons suddenly start to detach from the nucleus. Furthermore, the asymptotic behaviour of the radial density becomes $\exp(-a \sqrt{r})$, in contrast to the standard $\exp(-ar)$. Given these alternative definitions for the nuclear charge, and the potential importance of the long-range density behaviour in fitting the CS formula, the authors should provide a better

justification for their choice of Z_c . In their current results, it is not surprising that the Z_c and H-fitted parameters give such similar results as the RHF wave functions are qualitatively very similar at these two charges, i.e. the electrons are still bound at $Z_c=1.031177528$. It would therefore be interesting to compare the optimised CS parameters at both definitions $Z_c=1.031177528$ and $Z_c=0.828161008$ to understand the significance of the critical binding threshold.

Thank you for this comment. We chose $Z_c=1.031177528$ as this can hold a bound state, unlike the hydride ion $Z=1$ or $Z_c = 0.828161008$. However, we agree that we should justify our choice of Z_c and so this had been added at the beginning of the results section on HF wavefunctions (page 7) along with the appropriate references.

We have performed calculations using $Z_c=0.828161008$ with a 25-term wavefunction and this did not show any improvement to performance of the CS-type formula for the calculation of anions, in fact it was considerably worse. Two tables, V and VI, showing this have been included in the supplementary information under two sections:

- 1. Fitting to Critical Nuclear Charge Data, $Z_c= 0.828161008$*
- 2. Predicted Correlation Energies from $Z_c= 0.828161008$ Fit*

which shows that fitting to data calculated using this charge values produces very large errors in calculated correlation energies. In the text in the main paper we have added “This system [$Z_c= 0.828161008$] was also considered but did not provide an improved correlation energy formula, see electronic supplementary material, so is not discussed further.”

(6) Having identified the challenges of fitting the CS parameters, can the authors comment on potential ways to overcome these challenges?

To address the challenge of fitting the CS parameters the simplest way is to reduce the number of fitting parameters, thus reducing the flexibility of the function offering a more unique fitting solution. However, this is not advocated as a more appropriate route to improving correlation formulae is to go beyond the severe appropriations used in the formulation, and there are several papers in the literature that attempt to do this.

(7) In Supporting Information Table IV, the caption refers to “The percentage error between exact and calculated values is provided in brackets”, but I cannot find any brackets in the table.

The statement has been removed – thank you.